# Reaction Mechanisms of H_2_S Oxidation by Naphthoquinones

**DOI:** 10.3390/antiox13050619

**Published:** 2024-05-20

**Authors:** Kenneth R. Olson, Kasey J. Clear, Tsuyoshi Takata, Yan Gao, Zhilin Ma, Ella Pfaff, Anthony Travlos, Jennifer Luu, Katherine Wilson, Zachary Joseph, Ian Kyle, Stephen M. Kasko, Prentiss Jones Jr, Jon Fukuto, Ming Xian, Gang Wu, Karl D. Straub

**Affiliations:** 1Department of Physiology, Indiana University School of Medicine—South Bend Center, South Bend, IN 46617, USA; ttakata@nd.edu (T.T.); yangao@iu.edu (Y.G.); zma3@nd.edu (Z.M.); epfaff@nd.edu (E.P.); atravlos@nd.edu (A.T.); jluu@nd.edu (J.L.); kwilso22@nd.edu (K.W.); zjoseph@nd.edu (Z.J.); ikyle@nd.edu (I.K.); skasko@nd.edu (S.M.K.); 2Department of Biological Sciences, University of Notre Dame, Notre Dame, IN 46556, USA; 3Department of Chemistry and Biochemistry, Indiana University South Bend, South Bend, IN 46615, USA; kclear@iusb.edu; 4Toxicology Department, Western Michigan University Homer Stryker M.D. School of Medicine, Kalamazoo, MI 49007, USA; prentiss.jones@wmed.edu; 5Department of Chemistry, Sonoma State University, Rohnert Park, CA 94928, USA; fukuto@sonoma.edu; 6Department of Chemistry, Brown University, Providence, RI 02912, USA; ming_xian@brown.edu; 7Department of Internal Medicine, University of Texas-McGovern Medical School, Houston, TX 77030, USA; gang.wu@uth.tmc.edu; 8Central Arkansas Veteran’s Healthcare System, Little Rock, AR 72205, USA; kdstraub@gmail.com; 9Departments of Medicine and Biochemistry, University of Arkansas for Medical Sciences, Little Rock, AR 72202, USA

**Keywords:** reactive sulfur species, reactive oxygen species, antioxidants, naphthoquinones

## Abstract

1,4-naphthoquinones (NQs) catalytically oxidize H_2_S to per- and polysufides and sulfoxides, reduce oxygen to superoxide and hydrogen peroxide, and can form NQ-SH adducts through Michael addition. Here, we measured oxygen consumption and used sulfur-specific fluorophores, liquid chromatography tandem mass spectrometry (LC-MS/MS), and UV-Vis spectrometry to examine H_2_S oxidation by NQs with various substituent groups. In general, the order of H_2_S oxidization was DCNQ ~ juglone > 1,4-NQ > plumbagin >DMNQ ~ 2-MNQ > menadione, although this order varied somewhat depending on the experimental conditions. DMNQ does not form adducts with GSH or cysteine (Cys), yet it readily oxidizes H_2_S to polysulfides and sulfoxides. This suggests that H_2_S oxidation occurs at the carbonyl moiety and not at the quinoid 2 or 3 carbons, although the latter cannot be ruled out. We found little evidence from oxygen consumption studies or LC-MS/MS that NQs directly oxidize H_2_S_2–4_, and we propose that apparent reactions of NQs with inorganic polysulfides are due to H_2_S impurities in the polysulfides or an equilibrium between H_2_S and H_2_S_n_. Collectively, NQ oxidation of H_2_S forms a variety of products that include hydropersulfides, hydropolysulfides, sulfenylpolysulfides, sulfite, and thiosulfate, and some of these reactions may proceed until an insoluble S_8_ colloid is formed.

## 1. Introduction

Natural and synthetic 1,4-naphthoquinones (NQs) have been of interest for their general cytoprotective actions as well as specific anti-cancer, anti-inflammatory, anti-viral, and antiparasitic activities [1,2,3,4,5,6,7,8]. NQs catalytically undergo one- or two-electron oxidation–reduction reactions with endogenous reductants and oxygen as substrates and, as electrophiles, they form covalent adducts with select intracellular neutrophils such as thiols [9]. Many of their health benefits are attributed to these reactions. Redox cycling generates reactive oxygen species (ROS)—namely, superoxide (O_2_^•−^), which dismutes to hydrogen peroxide (H_2_O_2_) (the latter then activates Nrf2 and nuclear antioxidant response elements, or at higher concentrations it becomes a cytotoxic oxidant). NQs may also form adducts with low-molecular-weight thiols, amines, and Cys on regulatory proteins, and these may affect intracellular antioxidants and alter the protein function [3,9,10,11].

We recently demonstrated that NQs may use hydrogen sulfide (H_2_S) as the reductant and oxidize it to hydroper- and hydropolysulfides (H_2_S_n_ where n = 2–5), sulfite, and thiosulfate [12], or, in the presence of glutathione or Cys (GSH-S and Cys-S where S denotes the additional, reactive sulfur), to organic hydroper- and hydropolysulfides (GSH-S_n_ and Cys-S_n_, where n = 2–4) as well as GSH-S_2_OH [13]. We also observed that GSH, Cys, and propylamine readily formed adducts with NQs and these, in turn, variously affected H_2_S oxidation by NQs.

Conversely, Abiko et al. [14] reported that 9,10-phenanthraquinone (9,10-PQ) undergoes one-electron reduction reactions with inorganic polysulfides (Na_2_S_2_ and Na_2_S_4_) but not with Na_2_S (which forms H_2_S when dissolved). Furthermore, we failed to observe reactions between anthraquinone and H_2_S. These factors raise the question of whether there are fundamental differences between anthraquinones and naphthoquinones in their ability to redox cycle with small thiols.

While our previous studies were designed to examine individual products of NQ-catalyzed H_2_S oxidation, they were not well-suited to evaluating the overall efficacy of H_2_S oxidation by various NQs. Here, we evaluate these reactions in greater detail by monitoring the consumption of both H_2_S and oxygen by NQs under different conditions. We also use the oxygen consumption, liquid chromatography mass spectrometry (LC-MS/MS), and UV-Vis methods to examine the reactions between selected NQs and inorganic hydropersulfides: Na_2_S_2_, Na_2_S_3_, and Na_2_S_4_. We show that the general order of efficacy is DCNQ ~ juglone > 1,4-NQ > plumbagin ~ DMNQ ~ 2-MNQ ~ menadione, although this varies somewhat depending on the experimental conditions. While most NQs readily oxidize H_2_S, we can find little evidence that they oxidize H_2_S_2–4_, and we propose that apparent reactions with H_2_S_2–4_ result from either H_2_S impurities in the polysulfides or from the equilibrium between H_2_S and H_2_S_n_. We also suggest that NQ oxidation of H_2_S can initiate a series of reactions that lead to the formation of an insoluble S_8_ colloid.

## 2. Materials and Methods

### 2.1. H_2_S and Polysulfide Measurements in Buffer

Fluorophore experiments were performed in 96-well plates, and fluorescence was measured with a SpectraMax M5e plate reader (Molecular Devices, Sunnyvale, CA, USA). Compounds were pipetted into 96-well plates, and the plates were covered with tape to minimize H_2_S loss due to volatilization. Excitation/emission (Ex/Em) wavelengths were recorded as per the manufacturer’s recommendations, using 7-azido-4-methylcoumarin (AzMC, 365/450 nm) and 3′,6′-Di(O-thiosalicyl) fluorescein (SSP4, 482/515 nm). AzMC and SSP4 have been shown to have sufficient specificity relative to other sulfur compounds and reactive oxygen and nitrogen species (ROS and RNS, respectively) to effectively identify H_2_S by using AzMC, and to identify per- and polysulfides (H_2_S_2_ and H_2_S_n_ where n = 3–7, RS_n_H where n > 1, or RS_n_R = where n > 2) by using SSP4 [15,16,17]. As both AzMC and SSP4 are irreversible, they provide a cumulative record of H_2_S and polysulfide production, but they do not reflect cellular concentrations at any specific time.

### 2.2. Kinetics of H_2_S Metabolism by NQs

In the presence of oxygen, NQ-catalyzed oxidation of H_2_S produces a variety of products (e.g., polysulfides and thiosulfate) that cannot be analyzed by a single method. It is also likely that other products remain to be identified. With this in mind, we used the H_2_S-specific fluorophore AzMC to examine the time-dependent disappearance of H_2_S, to better understand the overall capacity of NQs to oxidize H_2_S. H_2_S, NQs, and other compounds of interest as needed were placed in five 96-well plates at room temperature and covered with tape to minimize H_2_S volatilization. The tape was removed at timed intervals, AzMC (25 μM) was added, and the wells were re-taped and counted on the plate reader 10 min later. A low buffer liquid/total well volume (150/300 μL) was employed to minimize the effect of oxygen depletion on H_2_S consumption and to promote autoxidation of the NQs to sustain the reaction. As shown in Appendix A, there was a slight decrease in fluorescence in the H_2_S-alone samples due to volatility and slow autoxidation of H_2_S, whereas fluorescence decreased at a faster rate in the presence of various NQs. The background fluorescence of the plates (typically ~600 counts, as evidenced by the nadir in fluorescence of the NQs that rapidly consumed H_2_S) was subtracted from all samples, and the results are expressed as the percent change in fluorescence (*f*) of the test sample relative to samples with H_2_S only, i.e., ((1 − (*f_NQH_*_2*S*_/*f_H2S_*))⋅100). This allowed us to minimize the effect of H_2_S volatilization and autoxidation, but statistical comparisons could not be drawn from these averages.

### 2.3. Oxygen Consumption by Naphthoquinones and H_2_S_n_

The oxygen tension was monitored in a stirred 1 mL water-jacketed chamber with a FireStingO_2_ oxygen sensing system (Pyroscience Sensor Technology, Aachen, Germany) using a non-oxygen-consuming 3 mm diameter OXROB10 fiberoptic probe at room temperature. The probe was calibrated with room air (21% O_2_) or nitrogen gas (0% O_2_). Compounds of interest were added at timed intervals and the percent of oxygen (100% equals room air) was measured every 0.1–0.3 s. The oxygen concentration in μM was estimated by multiplying the percent of oxygen by the solubility coefficient for oxygen in 300 mOsm saline at 20 °C (2.65 μM⋅L^−1^⋅% O_2_, i.e., for air-saturated buffer, 2.65⋅100 = 265 μmol oxygen). The oxygen consumption was calculated from the difference between the oxygen concentration immediately after adding the compounds of interest and the oxygen concentration at the response nadir.

The oxygen consumption was also used to measure the relative catalytic efficacy of different NQs, as this was the only parameter that could be measured in real-time. Although many traces appeared to be characteristic of a one- or two-component exponential decay, this could not be confirmed in log plots. As the next best approach, we calculated the net decrease in oxygen concentration over the initial 5 min after the addition of 300 μM H_2_S to 10 μM NQ (assuming this was relatively linear) and divided this by 5 to give the consumption in μmol⋅min^−1^. The rate of oxygen consumption for H_2_S alone was subtracted from the oxygen consumption by NQs that slowly consumed oxygen to correct for spontaneous oxidation. In additional experiments, the amount of oxygen consumed and the rate of oxygen consumption over the initial 2 min after H_2_S addition were examined after multiple additions of 100 μM H_2_S to 100 μM NQ, to evaluate the stability of the NQs.

The addition of polysulfides (Na_2_S_2–4_) produced a rapid decrease in oxygen followed by a slower decline. In these experiments, the amount of oxygen consumed in the rapid and slow phases was determined, to compare the catalytic efficacies of the NQs.

### 2.4. Preparation of Thiol and Amine Adducts

Thiol and amine adducts were prepared as described previously [13]. Briefly, 1 mM GSH or Cys was added to equimolar concentrations of NQs in open containers and incubated for 1 h at room temperature to allow adduct formation and autoxidation of the NQ adduct. H_2_S adducts were prepared by placing 1 mM H_2_S and 1 mM NQ in a closed container for 1 h, then opening the container for an additional hour to allow unreacted H_2_S to dissipate and the NQ-SH adduct to autoxidize. Propylamine and NQs (both 1 mM) were incubated overnight to allow adduct formation. The NQ adducts were then diluted to micromolar concentrations of the parent NQ for measurements of the H_2_S consumption, as described above.

### 2.5. Effects of Oxidant Scavengers on H_2_S Oxidation by DMNQ and DCNQ

The effects of Trolox, a probable scavenger of one electron oxidants, on H_2_S consumption by DMNQ and DCNQ were directly examined by incubating the NQ with or without Trolox, as described above. However, tempol, an SOD mimetic, directly consumed H_2_S (Appendix A) and its effects on the NQ-H_2_S reactions could not be accurately evaluated, so oxygen consumption was used as a surrogate.

### 2.6. LC-MS/MS Analyses of S_n_-NQ Reactions

Sulfur compounds containing Na_2_S, Na_2_S_2_, Na_2_S_3_, or Na_2_S_4_ were prepared as solutions in PBS at a final concentration of 0.1 mM. Sulfur solutions of 200 µL were separated for further experiments under both normoxic and hypoxic conditions, the latter produced by sparging PBS with 100% N_2_ for >20 min. For hypoxic reactions, 10 µM 1,4-NQ, DCNQ, or DMNQ was added, followed by 5 mM HPE-IAM and incubation at 37 °C for 20 min. For normoxic conditions, an additional step involved allowing the reaction to proceed at room temperature for 20 min prior to adding HPE-IAM. The samples were then analyzed using LC-MS/MS with selected ion monitoring (SIM).

LC-MS/MS analysis was conducted using a Waters Micromass Quattro Premier Triple Quadrupol Mass Spectrometer (Waters Corporation, Milford, MA, USA) coupled to the Waters Alliance 2795 LC system at Western Michigan University, Homer Stryker School of Medicine. Chromatographic separation was accomplished using a YMC-Triart C18 column with dimensions of 50 × 2.1 mm inner diameter, used with a mobile phase consisting of A (0.1% formic acid) and B (0.1% formic acid in methanol) with a flow rate of 0.2 mL/min at 40 °C, to which 10 μL was injected with a linear gradient (5–90% B, 0–4 min, and 90% B, 4–7 min). The mass spectrometer was operated using electrospray ionization in the positive ion mode with the capillary voltage set to 3500 V and drying gas set to 10.0 L/min at 350 °C. The HPE-IAM polysulfide adducts were detected as the [M + H]^+^ ion using their exact masses ± 0.002 *m*/*z*: S_1_ (389.153), S_2_ (421.125), S_3_ (453.097), S_4_ (485.069), S_5_ (517.041), S_6_ (549.013), S_7_ (580.985), HSO_3_ (260.059), and HS_2_O_3_ (292.031).

### 2.7. Formation of Colloidal Sulfur (S_8_)

H_2_S was added to 1,4-NQ in 100 mM air-saturated phosphate buffer, at pH 7.2 and 22 °C. The optical density (OD), due to the turbidity of the colloidal S_8_ sulfur formed, was monitored at 650 nm without or with bovine erythrocyte SOD or bovine liver catalase. GSH was added in one experiment to reduce the S_8_ colloid to H_2_S [18], and the decrease in OD confirmed the utility of our method.

### 2.8. Synthesis of 2-Hydroxy-3-methoxy-1,4-napathoquinone (HMNQ)

Sodium hydride (NaH, 0.29 g, 7.3 mmol, 60% in mineral oil) was added to an oven-dried round-bottomed flask containing 12 mL of anhydrous methanol. After bubbling subsided, 0.55 g (2.4 mmol) of 2,3-dichloro-1,4-naphthoquinone was added to the solution with stirring. The reaction mixture was heated to reflux for 5 h. After 5 h, 0.35 g additional NaH was mixed with 2 mL of anhydrous methanol and added to the reaction. Reflux was resumed for 1 h, and the reaction mixture was concentrated in vacuo. The residue was dissolved in water (20 mL), and 6 M H_2_SO_4_ was added to precipitate the product. The precipitate was isolated by vacuum filtration to give 0.39 g (80% yield) of HMNQ as an orange solid, which was used without further purification. mp 149–153 °C (lit 150–152 °C) [19]; ^1^H NMR (400 MHz, CDCl_3_) δ ppm: 8.10 (1 H, dd, J = 7.5, 1.3 Hz), 8.06 (1 H, dd, J = 7.5, 1.8 Hz), 7.74 (1 H, td, J = 7.6, 1.5 Hz), 7.69 (1 H, td, J = 7.6, 1.5 Hz), 4.20 (3 H, s), 4.12 (1 H, s).

### 2.9. Chemicals

SSP4 was purchased from Dojindo molecular Technologies Inc. (Rockville, MD, USA). All other chemicals were purchased from Sigma-Aldrich (St. Louis, MO, USA) or ThermoFisher Scientific (Grand Island, NY, USA). ‘H_2_S’ is used throughout to denote the total sulfide (sum of H_2_S + HS^−^) derived from Na_2_S; S^2−^ most likely does not exist under these conditions (May et al., 2018 [20]). Phosphate-buffered saline (PBS; in mM): 137 NaCl, 2.7 KCl, 8 Na_2_HPO_4_, and 2 NaH_2_PO_4_. The phosphate buffer for absorbance measurements (PB; in mM): 200 Na_2_PO_4_. pH was adjusted with 10 mM HCl or NaOH to pH 7.4. The chemical structures of NQs used in these experiments are inserted into Figure 1.

### 2.10. Statistical Analysis

Data were analyzed and graphed using QuatroPro X9 (Corel Corporation, Ottawa, ON, Canada) and SigmaPlot 13.0 (Systat Software, Inc., San Jose, CA, USA)). Statistical significance was determined with Student’s *t*-test or one-way ANOVA and the Holm–Sidak test for multiple comparisons, as appropriate, using SigmaStat 4.0 (Systat Software, San Jose, CA, USA). Results are given as the mean +/− SE; significance was assumed when *p* < 0.05.

## 3. Results

### 3.1. Kinetics of NQ-Catalyzed H_2_S Metabolism

The effects of different NQs on the extent and rate of H_2_S catabolism are shown in Figure 1. Within the first 30 min, nearly 80% of the H_2_S was removed by 10 μM 1,4-NQ, and it was essentially all cleared at 30 min by 30 μM 1,4-NQ (Figure 1A). Lower concentrations of 1,4-NQ consumed progressively less H_2_S (Figure 1B), showing that the effect of 1,4-NQ on HS consumption was concentration dependent. Juglone, with a hydroxyl on the 5-carbon, was as effective as 1,4-NQ (Figure 1C). Plumbagin, with a methyl in the quinoid carbon, was less efficacious than juglone (Figure 1D), and H_2_S consumption was further decreased with menadione, likely due to the loss of the 5-carbon hydroxyl (Figure 1E). The rate and extent of H_2_S removal then increased when the 2-carbon methyl group was replaced by the methoxy group in 2-methoxy-1,4-NQ (2-MNQ, Figure 1F), and it further increased with the addition of a second methoxy group at the 3-carbon in 2,3-dimethoxy-1,4-NQ (DMNQ, Figure 1G). Replacing the methoxy groups with chlorine in 2,3-dichloro-1,4-NQ (DCNQ, Figure 1H) consumed H_2_S faster than any of the other NQs. These results are similar to what we demonstrated for NQs with a single substitution on the quinoid group [13] and extend our findings to the more substituted DMNQ and DCNQ. HMNQ (2-hydroxy-3-methoxy-NQ) did not consume H_2_S, suggesting that the 2-hydroxy-3-methoxy-NQ has effects similar to the 2C hydroxyl in lawsone that does not react with H_2_S [13]. Our results also illustrate the utility of measuring H_2_S consumption to provide a more comprehensive appreciation of H_2_S metabolism by NQs.

### 3.2. Importance of C2 and C2 Quinoid Carbons in H_2_S Oxidation

As described in the introduction, there is uncertainty as to whether the initial H_2_S oxidation occurs at the carbonyl group or at the quinoid carbon atoms. Both potentially reactive C2 and C3 quinoid carbons of DCNQ and DMNQ are occupied with chlorine or methoxy groups, respectively, yet they consume H_2_S (Figure 1). A number of experiments were conducted in an attempt to identify the importance of substituent groups at these carbons for H_2_S oxidation. DMNQ was of particular interest as it does not form adducts with thiols such as GSH, yet it redox cycles as readily as menadione [21].

We have previously shown that the amount of polysulfide production (measured by SSP4 fluorescence) and the rate of oxygen consumption by a variety of NQs with open positions on the C2 or C3 quinoid carbon are dependent on the concentrations of both NQ and H_2_S [12,13]. Qualitatively similar results for polysulfide production (SSP4 fluorescence) were observed with DCNQ and DMNQ (Figure 2). SSP4 fluorescence was concentration-dependently increased by H_2_S when incubated with 10 μM or 30 μM DCNQ (Figure 2A,B) or DMNQ (Figure 2E,F), although production plateaued at 30 μM H_2_S with 10 μM DCNQ. With 300 μM H_2_S, DCNQ concentration-dependently increased SSP4 fluorescence up to 3–10 μM and appeared to decrease it thereafter (Figure 2C), whereas with DMNQ, fluorescence continuously increased (Figure 2G).

Both DCNQ (Figure 2D) and DMNQ (Figure 2H) concentration-dependently increased oxygen consumption when incubated with H_2_S. Complete oxidation of 300 μM H_2_S by DCNQ or DMNQ should consume all the ~200–265 μM oxygen in the chamber but clearly this was not the case for either NQ (Figure 2D,H). To determine if this was due to depletion of H_2_S or NQ, we measured oxygen consumption after sequential aliquots of either H_2_S or NQ. As shown in Figure 2I–K, oxygen consumption was not affected by additional DCNQ or DMNQ, whereas additions of H_2_S consumed progressively more oxygen until all oxygen was consumed. However, the oxygen tension never fell much below 40% after the initial treatment, even with high concentrations of NQs. This is equivalent to a decrease of ~160 μM oxygen, approximately half the amount of H_2_S that was added, suggesting that all the H_2_S was consumed. This was further examined by incubating 10 μM and 30 μM DCNQ with 300 μM H_2_S. As shown in Figure 2L, oxygen consumed by either 10 μM or 30 μM DCNQ did not fall much below 40%, and it was clear that the initial bolus of H_2_S was depleted prior to a second H_2_S addition.

These experiments show that H_2_S is the limiting factor, and they support our hypothesis that both DCNQ and DMNQ function as catalysts, as we have previously observed for other NQs [12,13]. The failure of the initial 300 μM H_2_S to consume all the oxygen could be partially explained by continual oxygen diffusion into the reaction chamber. However, this is unlikely given the relatively slow rate of reoxygenation compared to the rapid rate of oxygen depletion observed with DCNQ. Assuming a small contribution from ambient oxygen diffusion into the chamber, our experiments are more consistent with an oxygen:H_2_S consumption ratio of 1:2 in the initial reaction.

The rate of oxygen consumption was then used to measure the relative catalytic efficacy of different NQs, as this was the only parameter that could be measured in real-time. As shown in Figure 2M, juglone had the highest rate of oxygen consumption, at 7.2 ± 0.38 μmol∙min^−1^. The initial rate of oxygen consumption decreased in the following order: juglone > DCNQ > 1,4-NQ > plumbagin ~ DMNQ ~ 2-MNQ ~ menadione.

The low (≤30 μM) concentrations of NQs relative to H_2_S in the above experiments clearly demonstrated the catalytic properties of NQs, but these were too low to allow an examination of possible direct effects of NQs on oxygen consumption in air-saturated buffer, where the oxygen concentration was close to 265 μM. To examine these reactions, the oxygen consumption was monitored while 100 μM of selected NQs was incubated with multiple aliquots of 100 μM H_2_S. As shown in the traces of oxygen consumption (Appendix A), the effects of 1,4-NQ, juglone, plumbagin, and DMNQ were qualitatively similar, more oxygen was consumed by the first H_2_S treatment, and progressively less oxygen was consumed by the third and fourth H_2_S aliquots (summarized in Appendix A). The decrease in oxygen consumption between the first and second H_2_S was greatest for 1,4-NQ and juglone and progressively less for DMNQ and plumbagin. Conversely, half the amount of oxygen was consumed by the initial addition of H_2_S to DCNQ compared to the subsequent three additions, which were all similar. The amount of oxygen consumed during the first H_2_S treatment decreased in the order 1,4-NQ ~ juglone > plumbagin ~ DMNQ > DCNQ. Oxygen consumption by the second H_2_S addition was similar for all NQs, and by the third H_2_S addition, it was DCNQ > plumbagin > 1,4-NQ ~ DMNQ > juglone. By the fourth H_2_S addition, it was DCNQ > plumbagin > DMNQ ~ juglone ~ 1,4-NQ.

The decrease in oxygen tension in all these reactions could not be fit to a one- or two-component exponential curve, so the rate of change in oxygen tension over the initial 2 min was evaluated as a linear decrease (summarized in Appendix A). The rate of oxygen consumed after the first H_2_S decreased in the following order: juglone > DCNQ ~ 1,4-NQ > plumbagin ~ DMNQ. The rate of oxygen consumption by the second addition of H_2_S juglone decreased by 80% compared to the first, and the rates for 1,4-NQ and DMNQ also decreased by 58% and 43%, respectively. Conversely, the rate of oxygen consumption with plumbagin increased, while the rate for DCNQ remained near-constant. Adding H_2_S to DCNQ produced an abrupt, linear decrease in oxygen tension that quickly leveled off. Clearly, DCNQ reacts differently to DMNQ and 1,4-NQ, which is most likely due to a nucleophilic replacement of the chlorines by -SH [22,23,24].

If we assume that the simplest reaction would be 2H_2_S + O_2_ –> H_2_S_2_ + H_2_O_2_, then 50 μM of oxygen would be consumed by each addition of 100 μM H_2_S, i.e., a 1:2 O_2_:H_2_S ratio. Any increase in this ratio would suggest the production of sulfoxides or other sulfur oxidation products (SOPs), and a decrease would suggest that oxygen is consumed independent of H_2_S. As shown in Appendix A, the O_2_:H_2_S ratio for both 1,4-NQ and juglone was close to 1.5:2, whereas it was nearly equal to 1:2 for plumbagin and DMNQ, but only 0.5:2 for DCNQ. This suggests that the first addition of H_2_S to 1,4-NQ or juglone produced a substantial amount of SOP, whereas little SOP was initially produced by plumbagin or DMNQ. The 0.5:2 ratio for DCNQ suggests that half of the H_2_S consumed does not require oxygen, and this ratio is more consistent with the formation of a DCNQ-S-DCNQ dimer, which is also very efficacious in oxidizing H_2_S. With the second H_2_S aliquot, the O_2_:H_2_S ratios were close to 1:2 for all NQs. By the third and fourth H_2_S addition, the ratios for 1,4-NQ, juglone, and DMNQ were at or below 0.6:2, whereas those for plumbagin and DCNQ remained near 1:2. This suggests that less H_2_S is oxidized after multiple additions to 1,4-NQ, juglone, and DMNQ and that it may be consumed in the formation of various NQ-S_n_ and NQ-S_n_-NQ adducts or in anerobic reactions with other sulfur compounds, while there is little further adduct formation with plumbagin and DCNQ.

The oxygen consumption by 100 μM DCNQ and variable amounts of H_2_S was then examined to further characterize this reaction. As shown in Appendix A, increasing the H_2_S concentration from 50 μM to 300 μM produced a concentration-dependent increase in oxygen consumption, and in each instance, the decrease in oxygen was rapid and abruptly stopped without any indication of an exponential decline. Oxygen was also steadily consumed by H_2_S in the absence of DCNQ, and it was evident that towards the end of the experiment, more oxygen was consumed by 50 and 100 μM H_2_S alone than by H_2_S and DCNQ; similar results would likely have been observed at higher H_2_S concentrations if the sample time was extended. The late rise in oxygen tension was due to oxygen diffusing back into the chamber at a rate of just under 1 μmol⋅min^−1^. Collectively, these results suggest that H_2_S from 50 to 300 μM was completely consumed after the addition to 100 μM DCNQ, but that an initial fraction of this H_2_S did not result in oxygen consumption consistent with the formation of a DCNQ-S-DCNQ dimer or with nucleophilic replacement of chlorine by the sulfhydryl group from H_2_S. The formation of NQ-S adducts is being addressed in ongoing studies.

If quinone is directly involved in H_2_S oxidation, then we would not expect there to be a significant lag phase when the two are reacted, whereas, if another, intermediate species needs to be generated first (e.g., an NQ-S adduct), then a lag phase might be expected. To examine these possibilities, we measured the oxygen consumption for the initial 5–10 min after 300 μM H_2_S was added to 10 μM NQ. There was little evidence of a lag phase for any NQ (Appendix A).

### 3.3. Effects of GSH, Cys, and Propylamine Adducts on H_2_S Consumption by NQs

NQs with one or more unsubstituted quinoid C2 or C3 carbon may form adducts with GSH, Cys, or propylamine, which variously affect H_2_S oxidation to polysulfides [13]. A number of experiments were conducted to examine the effects of these adducts on H_2_S consumption with special reference to the stability of the Cl and O-methoxy groups of DCNQ and DMNQ.

As shown in Appendix A, GSH had a minimal effect on H_2_S consumption by 1,4-NQ, DMNQ, or DCNQ and slightly decreased H_2_S consumption by juglone. Conversely, GSH greatly increased H_2_S consumption by plumbagin and menadione and, to a lesser extent, by 2-MNQ. Cysteine decreased H_2_S consumption by 1,4-NQ, juglone, and DCNQ, slightly increased oxygen consumption by menadione, and had no effect on H_2_S consumption by plumbagin, 2-MNQ, or DMNQ. These results show that GSH has a minimal effect on H_2_S consumption by an NQ with open C2 and C3 carbons, decreases it when there is a hydroxyl on the benzene carbon, and increases it when one of the quinoid carbons is occupied with either a methyl or methoxy group. Conversely, Cys has no effect or decreases H_2_S consumption by all NQs except menadione. DMNQ with both C2 and C3 occupied with methoxy groups is refractory to both GSH and Cys, whereas Cys inhibits DCNQ, suggesting that Cl is a good leaving group, as discussed above.

Propylamine (PA) slowly forms adducts with NQs that have an unsubstituted C2 or C3 carbon, and this prevents further reactions of the NQs with GSH or Cys [13]. In the present experiments, PA and NQs were incubated overnight to provide sufficient time for adduct formation. As shown in Appendix A, 1,4-NQ-PA adducts consistently decreased H_2_S consumption compared to 1,4-NQ alone, and the inhibitory effect of a 1:1 NQ:PA ratio became progressively less as the concentration of the diluted adduct increased from 3 μM to 30 μM, whereas the inhibitory effect of a 1:5 1,4-NQ-PA ratio was unaffected by the concentration of the diluted adduct. Propylamine also inhibited H_2_S consumption by juglone (Appendix A), whereas it had no or a minimal effect on H_2_S consumption by menadione, plumbagin, or DMNQ (Appendix A). Propylamine also inhibited oxygen consumption by H_2_S and 1,4-NQ and by GSH and 1,4-NQ (Appendix A). These results suggest that substitutions on the C2 and/or C3 quinoid carbons affect the ability of NQs to form adducts with PA, and this, in turn, decreases the ability of propylamine to impact H_2_S oxidation. Collectively, the results also show that, as expected, H_2_S oxidation by DMNQ is essentially unaffected by GSH, Cys, or propylamine.

### 3.4. Effects of H_2_S Adducts on Polysulfide Production by DCNQ and DMNQ

The effects of potential H_2_S adducts on polysulfide production (SSP4 fluorescence) by DCNQ and DMNQ were examined because it was not practical to follow H_2_S levels with the AzMC fluorophore. As shown in Appendix A, DCNQ-SH adducts increased polysulfide production when incubated with 10 μM H_2_S, had a variable but small effect when incubated with 100 μM H_2_S, and only 10 μM DCNQ-SH decreased polysulfide production (best seen with 300 μM H_2_S in the reaction mixture). Conversely, all DMNQ-SH adducts increased polysulfide production when incubated with H_2_S. However, it should be noted that the maximum fluorescence obtained with the DMNQ-SH adducts was essentially the same, irrespective of whether the NQ was incubated with 10 μM, 100 μM, or 300 μM H_2_S. This suggests that the polysulfides in these samples were produced catalytically during the previous incubation of 1 mM H_2_S with 1 mM DMNQ and were not the result of subsequent incubation of H_2_S with the diluted adduct. Although not quite as clear, the responses of the DCNQ-SH adducts were similar. Collectively, these results suggest that even if H_2_S forms adducts with DCNQ or DMNQ, they do not substantially affect H_2_S oxidation to polysulfides.

### 3.5. Effects of ‘Antioxidants’ Trolox and Tempol on H_2_S and O_2_ Consumption by 1,4-NQ, DMNQ and DCNQ Reactions

The difficulties in characterizing the actions of ‘antioxidants’ notwithstanding [25], we examined the effects of Trolox and tempol on H_2_S metabolism by four NQs with different catalytic properties: 1,4-NQ, plumbagin, DMNQ, and DCNQ. Trolox did not affect H_2_S consumption by any NQ. However, it became evident in preliminary studies that tempol interfered with the reaction between H_2_S and AzMC, so oxygen consumption was monitored as a surrogate for H_2_S consumption. As shown in Appendix A, Trolox did not affect the oxygen consumption in the reactions of H_2_S with any NQ, which was consistent with the observations of H_2_S consumption under similar conditions. Conversely, tempol increased the consumption by all NQs except for DMNQ. Tempol dismutes superoxide and hydrogen peroxide and limits Fenton reactions [26]. Of these properties, only its SOD mimetic action appears to be consistent with its effect on H_2_S oxidation by NQs [12], presumably by scavenging superoxide and thereby favoring the oxygen-mediated oxidation of reduced NQs to their semiquinone [27].

### 3.6. Effects of SOD on NQ-Catalyzed H_2_S Metabolism

We have previously shown that SOD increases the oxygen consumption in NQ-catalyzed H_2_S oxidation [12]. However, the magnitude of this effect can be somewhat obscured by the efficacy of some NQs that rapidly consume H_2_S and by direct SOD-catalyzed oxidation of H_2_S [28]. To compensate for these, we decreased the concentration of the NQs to prevent 100% H_2_S consumption during the experiment, and we decreased the concentration of SOD to minimize its effect on H_2_S [28]. Under these conditions, SOD increased H_2_S consumption by all NQs (Appendix A). At 60 min, SOD increased H_2_S consumption by 1.5-fold for all NQs except for plumbagin, where it was increased by nearly three-fold, and DCNQ, where it was only increased by 0.9-fold (Appendix A). The SOD effect persisted for up to 120 min, although it decreased somewhat for the NQs where H_2_S consumption approached 100%, and it increased to 2.5-fold for menadione, which is the least efficacious in oxidizing H_2_S (Appendix A). The stimulatory effects of SOD are reported to be mediated by removing superoxide, a product of the one-electron oxidation of reduced NQH_2_ [27]. Our results suggest that this is a common feature of all reactions where H_2_S is oxidized by NQs.

### 3.7. Oxygen Consumption by NQ Reactions with Inorganic Hydropolysulfides

Abiko et al. [14] reported that selected quinones including 9,10-phenanthraquinone, pyrroloquinoline quinone, vitamin K_3_ (menadione), and coenzyme Q_10_ (CoQ_10_) redox cycle with oxygen to oxidize hydropersulfides (H_2_S_2–4_) but that they do not react with H_2_S. This contrasts with our previous and current observations that menadione, CoQ_10_, and a variety of other quinones and naphthoquinones oxidize H_2_S to polysulfides [12,13,29,30,31,32]. We also found very little evidence in prior studies—using K_2_S as a mixed polysulfide donor and detection with AzMC or SSP4—that these compounds oxidized polysufides. Here, we reexamined polysulfide metabolism by naphthoquinones using specific polysulfide salts—Na_2_S_2_, NaS_3_, and Na_2_S_4_—and we monitored the oxygen consumption as a real-time index of the redox cycling process. The results of these studies are shown in Figure 3 and summarized in Table 1.

The addition of Na_2_S_2_, Na_2_S_3_, or Na_2_S_4_ to room-air-equilibrated PBS produced a rapid decrease in oxygen over the initial 2 min and then a slower decrease that did not reach equilibrium 60 min later (Figure 3A). Approximately 2.5 times more oxygen was consumed during the fast component by Na_2_S_3_ and Na_2_S_4_ than by Na_2_S_2_, whereas there was no difference between them during the slow phase.

In general, all NQs increased the total oxygen consumption to approximately the same extent (~100 μmoles). However, the fraction consumed during the fast component increased as the number of sulfur atoms increased and the fraction of oxygen consumed by the slow component reciprocally decreased (Figure 3B,C). The fast component of oxygen consumption with Na_2_S_2_ or Na_2_S_3_ was increased by NQs in the following order: juglone > DCNQ ~DMNQ > 1,4-NQ. With Na_2_S_4_, there was no difference between juglone, DCNQ, and DMNQ, and all were greater than 1,4-NQ. All NQs increased the slow component, and the slow component of oxygen consumption by 1,4-NQ with Na_2_S_3_ or Na_2_S_4_ was greater than the other NQs, presumably because 1,4-NQ’s fast component was lesser.

Oxygen consumption with juglone, DCNQ, or DMNQ reached a nadir before 75 min, whereas with 1,4-NQ, oxygen was continuously consumed, albeit at a slower rate, and in several instances (shown, for example, with Na_2_S_2_ and Na_2_S_3_), there appeared to be an additional component. In no instance did oxygen fall below ~45%, suggesting that one of the reactants, likely a sulfur species, was the limiting factor. This was similar to the result observed for H_2_S (see Section 3.2 and Figure 2I–K), although the molar ratio of oxygen consumed to moles of H_2_S_n_ added was around 1:2.5. When correcting for the moles of sulfur in the different polysulfides, the O_2_:S ratios would be 1:5, 1:17, and 1:10 for Na_2_S_2–4_, respectively. Clearly, the amount of oxygen consumed was not dependent on the total sulfur but on some fraction that was relatively constant and independent of the total moles of sulfur. When using AzMC to estimate the amount of H_2_S in the hydropersulfides, it appeared that there was ~250 μM of H_2_S as a contaminant in all three hydropolysulfides (Figure 3D). This suggests that the source of much, if not all, of the oxygen consumed in the NQ-S_2–4_ reactions was the oxidation of H_2_S. However, unlike hydropolysulfides, there was no fast oxygen consumption with H_2_S alone or H_2_S with NQs (cf. Appendix A). This raises the possibility that hydropolysulfides specifically contribute to oxygen consumption through mechanisms in lieu of, or in addition to, direct redox cycling reactions, or that other factors, e.g., metal contaminants, are involved.

We then reversed the order of Na_2_S_n_-NQ addition to further examine the fast component of oxygen consumption and to determine if sulfur was the limiting factor. The addition of Na_2_S_3_ to the chamber produced the characteristic rapid decrease in oxygen, which was further decreased by DCNQ and by two subsequent additions of Na_2_S_3_ (Figure 4A). This suggests that Na_2_S_3_ is the limiting factor, as is Na_2_S (Figure 2I). It was also evident that when 10 μM DCNQ was added after Na_2_S_3_, there was no rapid drop in oxygen and, instead, oxygen decreased exponentially, essentially in a pattern identical to that observed when 10 μM DCNQ was added to 300 μM H_2_S (Figure 2D). This raises the question, how much of the oxygen consumed in a DCNQ-Na_2_S_n_ reaction is due to DCNQ reacting with H_2_S, the latter being present as a contaminant or produced in an equilibrium reaction with H_2_S_n_? To answer this question, we measured the oxygen consumption by DCNQ and juglone with Na_2_S_1–4_ with or without the H_2_S scavenging compounds SS20 and SS16 [33,34]. As shown in Figure 4B,C, SS20 concentration-dependently inhibited oxygen consumption by Na_2_S and Na_2_S_2_ reactions with DCNQ, and 250 μM SS20 inhibited reactions between DCNQ, Na_2_S_3_, and Na_2_S_4_. Both SS20 and SS16 inhibited oxygen consumption by Na_2_S_4_ and juglone. These results suggest that H_2_S accounts for most, if not all, of the oxygen consumed in reactions between H_2_S_2–4_ and NQs.

### 3.8. LC-MS/MS Analysis of NQ Reactions with Inorganic Hydropolysulfides

We then identified the products formed in NQ-H_2_S_n_ reactions with LC-MS/MS. Individual sulfur compounds (detected) produced by 20 min incubation of 100 μM Na_2_S Na_2_S_2_, Na_2_S_3_, or Na_2_S_4_ (source) without or with 10 μM 1,4-NQ, 10 μM DCNQ, or 10 μM DMNQ in either 21% or <1% O_2_ are shown in Figure 5A. Figure 5B shows the cumulative area under the curve for all sulfur species detected as a function of the sulfur species added (source). All Na_2_S_n_ salts produced polysulfide species from S_1_ to S_5_ as well as sulfite (H_2_SO_3_) and thiosulfate (H_2_S_2_O_3_) when dissolved, indicative of their general instability.

In general, without NQs, the relative AUC of the smallest sulfur-source and sulfur-detected species (Na_2_S_1–2_) was greater in a low (<1%) rather than a high (21%) oxygen buffer, whereas these differences became less pronounced as the number of S atoms in both the source and detected polysulfides increased. This suggests that smaller species are more prone to autoxidation. Sulfite and thiosulfate were more prevalent in 21% oxygen, suggesting some autoxidation of H_2_S_1–4_.

The H_2_S AUC for all source Na_2_S_1–4_ incubated in 21% oxygen was decreased by 1,4-NQ and almost eliminated by DCNQ, but it was only minimally affected by DMNQ. As the Na_2_S_1–4_ samples were incubated with NQs for 20 min, these responses likely reflect the oxidation of H_2_S, and the order of efficacy DCNQ > 1,4-NQ > DMNQ is consistent with that which can be observed for H_2_S consumption in Figure 1. A similar order was observed for samples incubated in <1% oxygen, albeit with less efficacy. H_2_S_6_ and H_2_S_7_ were not detected with H_2_S as the source. These results show that H_2_S is consumed by NQs irrespective of the source NaS_n_, and that there is little evidence for net production of H_2_S from polysulfides.

With Na_2_S as the source, and in 21% oxygen, DCNQ significantly decreased the formation of H_2_S_2_ and H_2_S_3_ but it did not affect H_2_S_4_ or H_2_S_5_, and 1,4-NQ and DMNQ did not affect H_2_S_2–5_. Conversely, in <1% oxygen, DCNQ increased H_2_S_2–5_, and 1,4-NQ increased H_2_S_2,3_. H_2_S_6_ and H_2_S_7_ were not detected in these samples. These results suggest that polysulfides are produced by NQ oxidation of H_2_S, but that in the presence of 21% oxygen, they are further oxidized to species other than polysulfides or sulfoxides.

In 21% oxygen, 1,4-NQ and DCNQ progressively decreased H_2_S_2_ and H_2_S_3_ formation irrespective of whether the source was Na_2_S_2_, Na_2_S_3_, or Na_2_S_4_, while DMNQ had no effect. The effects on H_2_S_4_ were less consistent. 1,4-NQ decreased H_2_S_4_ production from Na_2_S_3_ and Na_2_S_4_, whereas DCNQ decreased H_2_S_4_ production from Na_2_S_2_ and Na_2_S_3_. H_2_S_5_ production was increased by 1,4-NQ with Na_2_S_2_ and decreased by DCNQ with Na_2_S_3_ and Na_2_S_4_, while the production of H_2_S_6_ was generally similar to that of H_2_S_5_. In <1% oxygen, 1,4-NQ and DCNQ progressively decreased H_2_S_2_ and H_2_S_3_ production from Na_2_S_2_ and Na_2_S_3_, and DCNQ progressively decreased H_2_S_4_ production from Na_2_S_4_. The effects of 1,4-NQ and DCNQ on H_2_S_4_ production were variable, increasing production from Na_2_S_2_ and Na_2_S_3_ and decreasing it from Na_2_S_4_. This appeared to be a transition point, as both NQs increased H_2_S_5_ and H_2_S_6_ production from all three polysulfides.

Sulfite production from all Na_2_S_n_ was decreased by 1,4-NQ and DCNQ in 21% oxygen, along with that from Na_2_S and Na_2_S_2_ in <1% oxygen. Thiosulfate production was variously affected by NQs, Na_2_S_n_, and oxygen tension with no consistent pattern.

We were able to detect two sulfenylated polysulfides, HS_4_OH and HS_5_OH, in these reactions (Appendix A). In general, approximately twice as much HS_4_OH was produced compared to HS_5_OH, twice as much sulfenylated polysulfide was produced in 21% oxygen compared to <1% oxygen, and the amounts of HS_4_OH and HS_5_OH formed were directly proportional to the number of sulfur atoms in the source Na_2_S_1–4_, irrespective of the absence or presence of an NQ or the specific NQ employed. DCNQ produced more sulfenylated polysulfides than any other NQ regardless of oxygen tension. This was followed by 1,4-NQ, while DMNQ was generally ineffective. The effects of different NQs on Na_2_S_4_ and Na_2_S_5_ formation were most noticeable when Na_2_S was the source of sulfur. Sulfenylated species accounted for less than 10% of the total, and most were under 3%.

Collectively, these results suggest that in 21% oxygen, 1,4-NQ and, to a greater extent, DCNQ produce a net decrease in all sulfur species detected by LC-MS/MS irrespective of the degree of catenation of the initial (source) sulfur. The modest or nil effect of DMNQ is likely due to the generally slow reactivity of this NQ. The net consumption of sulfur compounds by 1,4-NQ and DCNQ is supported by comparing the sums of all AUCs for individual sulfur species (Figure 5B). It is evident that there is a decrease in the AUC when samples are incubated with 1,4-NQ and DCNQ, especially when samples are incubated in 21% O_2_. We hypothesize that among the undetected sulfur species is S_8_, as is examined in the following section.

### 3.9. Formation of S_8_ Colloid by 1,4-NQ Oxidation of Na_2_S and the Effects of SOD and Catalase

We noticed in a number of reactions between NQs and H_2_S that the solution became cloudy, which we interpreted as the result of colloidal sulfur (S_8_) formation. These reactions were then examined in more detail by comparing the optical density at 263 nm (OD_263_), purportedly the absorption maximum of soluble S_8_ [35], with the OD at 650 nm (OD_650_), where S_8_ does not absorb light but colloid formation will increase turbidity.

The spectra of 1,4-NQ before and after the addition of H_2_S and the effects of filtering, centrifuging, and resuspending the precipitate are shown in Figure 6A–C, where Figure 6A shows the spectrum from 190 m to 700 nm and Figure 6B,C show OD_263_ and OD_650_, respectively. 1,4-NQ, soluble S_8_, colloid, and presumably other sulfur compounds contribute to OD_263_. OD_263_ is approximately halved by filtration of centrifugation, and some of the centrifuged colloid can be resuspended. OD_650_ appears to be solely due to the colloid, as there is essentially no absorbance of light by 1,4-NQ or by the 1,4-NQ-H_2_S reaction product after filtration or centrifugation, whereas much of the colloid can be recovered from the centrifuged pellet.

K_2_S spontaneously oxidizes to form polysulfides. To examine if a colloid is also produced, we added K_2_S to a buffer either equilibrated with room air (21% O_2_) or sparged with N_2_ for 20 min (<1% O_2_), and we monitored OD_263_ and OD_650_ (Figure 6D,E, respectively). OD_263 nm_ decreased relatively quickly (and exponentially) when K_2_S was dissolved in <1% O_2_, whereas when it was dissolved in 21% O_2_, the decrease was slower and linear. The rate of decrease in OD_263_ in <1% O_2_ samples with SOD paralleled that of K_2_S in 21% O_2_ samples with or without SOD. Colloid formation (OD_650 nm_) was considerably greater in <1% O_2_ than in 21% O_2_, and SOD inhibited colloid formation to the extent that OD was similar to that in 21% O_2_. In 21% O_2_, SOD produced a further decrease in OD_650 nm_. These results validate the use of OD_650_ as an index of colloid formation and demonstrate that this reaction is oxygen sensitive and inhibited by SOD. The rapid decrease in OD_263 nm_ and increase in OD_650 nm_ when there is low oxygen may reflect anerobic conversion of polysulfides to an S_8_ colloid.

We then examined colloid formation in H_2_S oxidation by 1,4-NQ. As shown in Figure 6F,G, the incubation of Na_2_S and 1,4-NQ produced both soluble and colloidal sulfur in 21% O_2_, whereas there was less soluble sulfur and minimal colloid produced in <1% O_2_. The addition of 1 mM GSH had minimal effects on soluble sulfur, but it rapidly depleted the colloid formed in 21% O_2_. SOD increased soluble and colloidal sulfur production, whereas catalase was generally ineffective (Figure 6H,I). These results support our hypothesis that some, if not all, of the unaccounted sulfur in the LC-MS/MS experiments (Section 3.8) is colloidal sulfur. Our results also show that while oxygen is required for colloid formation in H_2_S-NQ reactions, it paradoxically decreases the colloid spontaneously formed from polysulfides.

## 4. Discussion

### 4.1. Background

The biological activities of naphthoquinones (NQs) are achieved through their ability to both redox cycle and function as electrophiles. In previous studies, we demonstrated that the oxidation of H_2_S by NQs is a catalytic process that reduces NQs, consumes oxygen, and generates a variety of inorganic polysulfides and sulfoxides (e.g., thiosulfate and sulfite), or their organic congeners when in the presence of other small thiols such as GSH and Cys. The efficacy of these reactions was also NQ-specific. Furthermore, certain NQs formed adducts with small thiols (GSH and Cys), or amines (propylamine and albumin) that, in turn, affected H_2_S oxidation [12,13]. In the present experiments, we took a different approach by examining the NQ-catalyzed consumption of H_2_S and oxygen. We then extended this work to include an examination of the reactions between NQs and inorganic hydroper- and hydropolysulfides (H_2_S_n_, where n = 2–4). Our initial studies helped to improve our understanding of the catalytic process, which we found to be less dependent than expected on the identities of the individual products that were ultimately formed and the rates of their formation. Further work with NQ-polysulfide reactions provided a better picture of the products and the overall reactions that were involved.

### 4.2. General Features of NQ-Catalyzed H_2_S Consumption

Based on H_2_S consumption (Figure 1), the overall efficacy of NQ-H_2_S reactions produced by the compounds used in this study appeared to be as follows: DCNQ > 1,4-NQ ~ juglone > plumbagin ~ DMNQ > 2-MNQ > menadione. However, the rate of oxygen consumption, which was the only parameter that could be measured in real-time, showed that 10 μM juglone had the highest initial rate of oxygen consumption, at 7.2 ± 0.38 μmol⋅min^−1^, and that the rate of consumption decreased in the following order: juglone > DCNQ > 1,4-NQ > plumbagin ~ DMNQ ~ 2-MNQ ~ menadione (Figure 2M). The ratio of oxygen to H_2_S consumed by 10 μM NQs at the initial nadir (cf. Figure 2D,H) was 0.5:1, suggesting that the overall net reaction can be written as the production of hydrogen persulfide and hydrogen peroxide (Equation (1)),
O_2_ + 2H_2_S –> H_2_S_2_ + H_2_O_2_,(1)
and that more complex polysulfides and sulfoxides were produced in subsequent reactions. Therefore, even 10 μM menadione with an oxygen consumption rate of 0.55 ± 0.04 μmol⋅min^−1^ would consume over 1 μmol of H_2_S each minute.

Incubation of 100 μM 1,4-NQ, juglone, plumbagin, DMNQ, or DCNQ with four or five consecutive doses of 100 μM H_2_S showed that the magnitude and rate of oxygen consumption varied with the dose and NQ (Appendix A). The initial dose of both 1,4-NQ and juglone consumed more oxygen than predicted by Equation (1), and DCNQ consumed considerably less. This suggests that NQs undergo reactions with H_2_S that are in addition to its catalytic activity. It is also likely that these reactions will affect the extent and rate of subsequent reactions, as demonstrated in Appendix A. 

It is clear that substituent groups on the benzene and quinoid rings profoundly affect H_2_S consumption. One OH group on the benzene ring (juglone) or two Cl on the quinoid (DCNQ) enhances H_2_S consumption compared to 1,4-NQ, whereas one or two methoxy groups (2-MNQ and DMNQ, respectively) or one methyl group (menadione) impedes it. Furthermore, adding a methyl group to the juglone quinoid, as in plumbagin, decreases H_2_S consumption. Lawsone, with a single quinoid hydroxyl, does not appear to react at all with H_2_S [12], nor does HMNQ with a quinoid hydroxyl and methoxy group. These reactions are generally consistent with the available one- and two-electron reduction potentials [12], although it is less clear if the efficacy of substituent groups specifically affects all or part of the NQ reduction/H_2_S oxidation reaction, autoxidation of the reduced NQ, or other unidentified reactions. There could also be steric effects, especially with substituent groups on both the quinoid 2 and 3 carbons, although these seem to affect reactions with GSH, Cys, and amines more than reactions with H_2_S. The possible reaction mechanisms that were examined in this study are illustrated in Figure 7 and described in the following sections. Fully protonated sulfur species are shown for clarity although the degree of protonation varies with individual species’ pKa.

### 4.3. H_2_S Oxidation and NQ Reduction: Possible Reactions

There are a number of possible reactions for NQ-mediated H_2_S oxidization, with two being the most notable. In the first scheme (Figure 7A), proposed by Tarumi et al. [36], a one-electron reaction initially oxidizes H_2_S (more likely the hydrosulfide anion, HS^−^) to a hydrosulfide radical (HS^•^) and reduces the NQ to a semiquinone radical (NQH^•−^, Equation (2)). A second one-electron reaction then reduces a second H_2_S to a hydrosulfide radical and fully reduces the semiquinone to NQH_2_ (Equation (3)). Two hydrosulfide radicals then combine to form the persulfide, H_2_S_2_ (Equation (4)).
NQ + H_2_S –> NQH^•−^ + HS^•^,(2)
NQH^•−^ + H_2_S –> NQH_2_ + HS^•^,(3)
HS^•^ + HS^•^ –> H_2_S_2_.(4)

Alternatively, the hydrosulfide radical could react with H_2_S to produce a persulfide radical (Equation (5)), which would then react with molecular oxygen to produce the persulfide and superoxide (Figure 7H; Equation (6)),
HS^•^ + H_2_S –> H_2_S_2_^•^,(5)
H_2_S_2_^•^ + O_2_ –> H_2_S_2_ + O_2_^•−^.(6)

The second scheme described by Perlinger et al. [37] for juglone involves H_2_S oxidation on the quinoid carbon by Michael addition, which produces an NQ-SH adduct without concomitant production of a semiquinone (Equation (7) and Figure 7B),
NQ + HS^−^ –> NQ^−^-SH.(7)

Perlinger et al. [37] also showed that at an elevated pH, a hydrosulfide radical (HS^•^) could react with juglone to form the NQ-SH adduct (Equation (8) and Figure 7C),
NQ + HS^•^ –> NQ^•^-SH,(8)
although this is unlikely in our experiments at pH 7.4.

If an NQ-SH adduct was formed, this would have to be followed by reaction of the adduct with a second H_2_S-, which would fully reduce the NQ to NQH_2_ and liberate the persulfide (Equation (9)), as shown in Figure 7B,
NQ-SH + H_2_S +H^+^ –> NQH_2_ + H_2_S_2_.(9)

Our experiments suggest that the Perlinger scheme is less likely. We showed that H_2_S was oxidized by DMNQ. The methoxy groups on DMNQ are impervious to adduct formation by either GSH or Cys, yet DMNQ readily redox cycles [21]. If we can assume that the quinoid C2 and C3 similarly will not form an adduct with H_2_S by displacing the methoxy groups, then the most plausible scenario for H_2_S oxidation involves the carbonyl oxygen, as depicted in Figure 7A.

### 4.4. NQ Autoxidation: Possible Reactions

A two-electron reaction between NQH_2_ and oxygen is spin restricted and unlikely to occur [27]. While a one-electron oxidation of the semiquinone, NQ^•^, is relatively facile, the preceding one-electron reaction between oxygen and NQH_2_ is not thermodynamically favorable, and this appears to be the limiting process in NQ reoxidation [27]. There are two scenarios for the initial generation of NQ^•^, a one-electron oxidation by oxygen that also produces superoxide (Equation (10) and Figure 7D), or a comproportionation reaction between NQ and NQH_2_ (Equation (11) and Figure 7E). Although Equation (10) is not favored, it can be enhanced by the addition of SOD and removal of the superoxide product [27].
NQH_2_ + O_2_ –> NQH^•^ + O_2_^•−^ + H^+^.(10)
NQ + NQH_2_ <–> 2NQ^•^ + H^+^,(11)

We previously used SOD in an attempt to identify the relative contributions of oxygen-mediated oxidation and comproportionation in autoxidation reactions [12]. However, we did not correct for H_2_S oxidation by SOD [28]. Here, we show that, when we corrected for this, it was evident that SOD had essentially the same effect on H_2_S oxidation by 1,4-NQ, juglone, menadione, and DMNQ, whereas H_2_S oxidation was somewhat noted, and was transiently increased by plumbagin and 2-MNQ and decreased by DCNQ (Appendix A). This suggests that there are comparable levels of oxygen-mediated oxidation for all reduced NQs. This also suggests that the different rates of H_2_S oxidation by NQs are mostly due to the initial H_2_S-oxidation, NQ-reduction reaction and not subsequent reoxidation of NQs.

It is generally assumed that SOD affects the one-electron oxidation of the fully reduced NQs and that this favors the oxidation of NQs that do not readily comproportionate (reaction D in Figure 7) [12,27]. Although SOD may also affect the one-electron oxidation of the semiquinone (reaction F in Figure 7), this reaction is already favored [27], and SOD will likely have little effect. We can also rule out the oxidation of semiquinones by superoxide (reaction G in Figure 7), as SOD will be expected to inhibit this reaction. The decreased effect of SOD on the H_2_S-DCNQ reaction (Appendix A) can be explained by the oxygen-independent formation of DCNQ-S adducts in the initial reaction (Section 4.3), which decreases the amount of oxygen available for the redox cycling of DCNQ.

### 4.5. Formation of Catenated Polysulfides

Our experiments indicate that the progressive catenation of polysulfides produced by NQ-catalyzed H_2_S oxidation does not account for a substantial fraction of the total oxygen consumed in these reactions. Furthermore, some colloidal sulfur formation appears to be favored in anoxic environments. This suggests a series of aerobic and anaerobic reactions involving sulfur radicals, comproportionation–disproportionation reactions, and sulfur exchange. In the first instance, the hydrosulfide radical produced by NQ oxidation (Equation (2)) would react with H_2_S to produce a hydropersulfide radical (Equation (5)). The hydropersulfide radical could react with a second H_2_S to produce a hydrotrisulfide radical (Equation (12)), which could react with another H_2_S to produce a hydrotetrapersulfide radical, and so on (Equation (13)),
H_2_S_2_^•^ + H_2_S –> H_2_S_3_^•^ + 2H^+^,(12)
H_2_S_n_^•^ + H_2_S –> H_2_S_n+1_^•^ + 2H^+^,(13)
or until it is oxidized by molecular oxygen, as in Equation (6). Two hydropersulfide radicals could also combine to produce a hydrotetrapersulfide (Equation (14)),
HS_2_^•^ + HS_2_^•^ –> H_2_S_4_.(14)

A variety of catenated hydropersulfides can also be generated by comproportionation (Equation (15)) and disproportionation (Equation (16)) reactions,
2H_2_S_2_ –> H_2_S_3_ + H_2_S,(15)
H_2_S_3_ + H_2_S –> 2H_2_S_2_,(16)
or through hydration/dehydration reactions. Here, a hydropersulfide is hydrolyzed to a hydrosulfoxide (Equation (17)),
H_2_S_2_ + H_2_O –> HSOH + H_2_S,(17)
two of which are then condensed to a hydrodisulfoxide (Equation (18)).
2HSOH –> HSSOH + H_2_O,(18)
which can continue to form longer-chain polysulfoxides (Equation (19)),
HSOH + HS_n_OH –> HS_n+1_OH + H_2_O,(19)

Progressively longer polysulfides can also be produced through S-S exchange reactions (Equations (20)–(22)),
H_2_S_2_ + H_2_S_2_ –> H_2_S_3_ + H_2_S,(20)
H_2_S_2_ + H_2_S_n_ –> H_2_S_n+1_ + H_2_S,(21)
H_2_S_n_ + H_2_S_y_ –> H_2_S_n+y-1_ + H_2_S,(22)
where n + y ≤ 9. These S-S exchange reactions increase the number of sulfur atoms in the polysulfide, and they produce H_2_S. It is likely that H_2_S is readily oxidized by NQs, further driving the reaction to the right. We have relatively little evidence that hydroper- or hydropolysulfides are further metabolized by NQs. With the exception of disproportionation, the above reactions could continue up to S_8_, at which point the octasulfide will cyclize and become insoluble, i.e., colloidal sulfur.

## Figures and Tables

**Figure 1 antioxidants-13-00619-f001:**
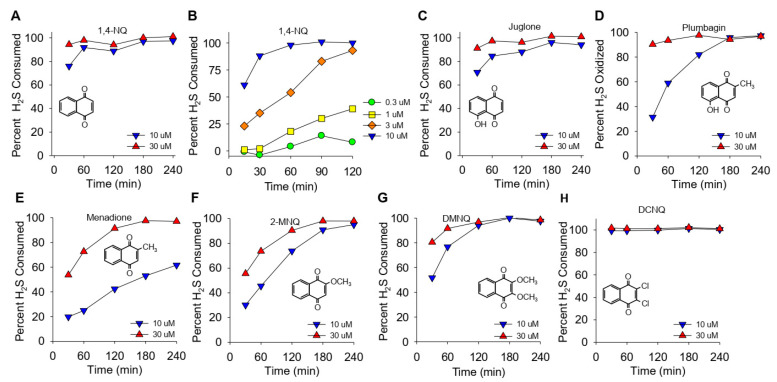
(**A**–**H**) Kinetics of NQ-catalyzed disappearance of H_2_S (AzMC fluorescence). Here, 10 or 30 μM NQs (species indicated in panels) was incubated with 100 μM H_2_S in taped well plates. AzMC (25 μM) was added after 30, 60, 90, 120, 180, or 240 min, and the samples were re-taped and counted at 10 min intervals. Values are expressed as the percent of H_2_S removed from the solution by NQs. Inserts show the NQ speciesstructure.

**Figure 2 antioxidants-13-00619-f002:**
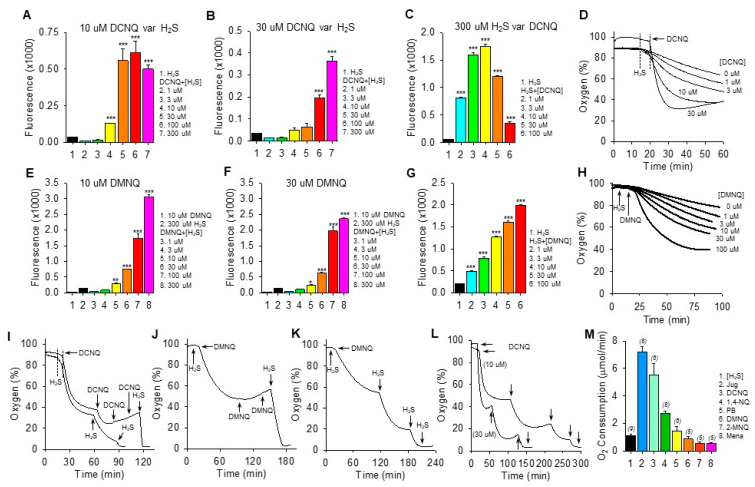
H_2_S oxidation to polysulfides by 2,3-dichloro-1,4-naphthoquinone (DCNQ) and 2,3-dimethoxy-1,4-naphthoquinone (DMNQ). H_2_S (300 μM) alone had a minimal effect on SSP4 fluorescence, whereas it concentration-dependently increased polysulfides (SSP4 fluorescence) when incubated with either 10 μM DCNQ or DMNQ ((**A**,**E**), respectively) or 30 μM DCNQ or DMNQ ((**B**,**F**), respectively). (**C**) DCNQ incubation with 300 μM H_2_S concentration-dependently increased polysulfide production up to 10 μM DCNQ and decreased it thereafter. (**G**) Polysulfide production from 300 μM H_2_S was concentration-dependently increased by DMNQ from 1 μM to 100 μM. Oxygen consumption by 300 μM H_2_S was concentration-dependently increased by DCNQ (**D**) or DMNQ (**H**). (**I**–**K**) Consecutive additions of 300 μM H_2_S to DCNQ or DMNQ further increased oxygen consumption, whereas additions of 10 μM DCNQ or DMNQ did not. (**L**) Effects of repetitive additions of 300 μM H_2_S (arrows) on oxygen consumption by 10 μM or 30 μM DCNQ. (**M**) Oxygen consumption in the initial five min after the addition of 300 μM H_2_S to 10 μM NQ. SSP4 values (**A**–**C**,**E**–**G**) summarize the effects after 100 min of reaction, mean + SE, n = 4 wells; *, *p* < 0.05, **, *p* < 0.01, ***, *p* < 0.001 vs. H_2_S without NQs. Oxygen consumption results presented as typical traces (**D**,**H**–**L**) or as mean + SE (*n*), as indicated in the figure, were all significantly different from each other (*p* < 0.01) except for DMNQ, 2-MNQ, and Mena.

**Figure 3 antioxidants-13-00619-f003:**
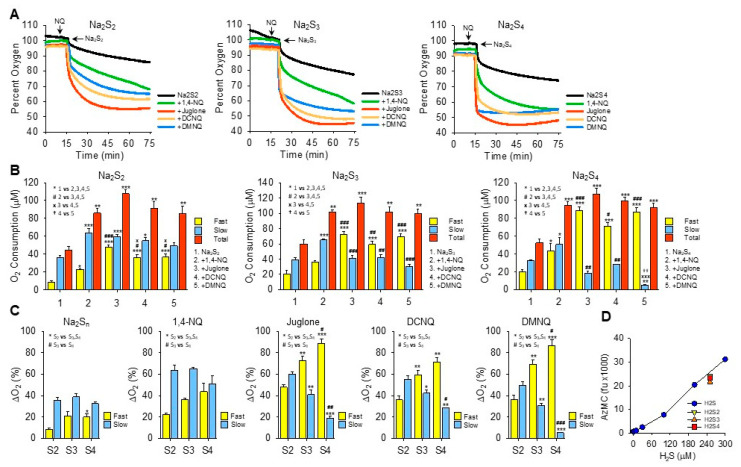
Effects of NQs on oxygen consumption by inorganic hydropersulfides. (**A**) Typical traces of oxygen consumption by 250 μM Na_2_S_2_, Na_2_S_3_, or Na_2_S_4_ alone (black) or after the addition of 10 μM 1,4-NQ (green), juglone (red), DCNQ (orange), or DMNQ (blue). (**B**) Summary of oxygen consumption during the fast and slow phases and total (fast plus slow) oxygen consumed by Na_2_S_2–4_ and 10 μM NQs. Mean + SE, n = 3 replicates, symbols in inset show statistical comparisons, where 1, 2 or 3 symbols represent *p* ≤ 0.05, *p* ≤ 0.01, or *p* ≤ 0.001, respectively. (**C**) Comparison of fast and slow O_2_ consumption components from (**B**) based on the number of sulfur atoms. (**D**) AzMC fluorescence from 250 μM Na_2_S_2–4_ (yellow, orange, and red symbols, respectively) compared to H_2_S standard curve (blue symbols). Estimated H_2_S results from NaS_2–4_ are 237 μM, 211 μM, and 227 μM, respectively; mean + SE, n = 4. Na_2_S_3_ is significantly less than Na_2_S_2_ (*p* = 0.002) or Na_2_S_4_ (*p* = 0.019).

**Figure 4 antioxidants-13-00619-f004:**
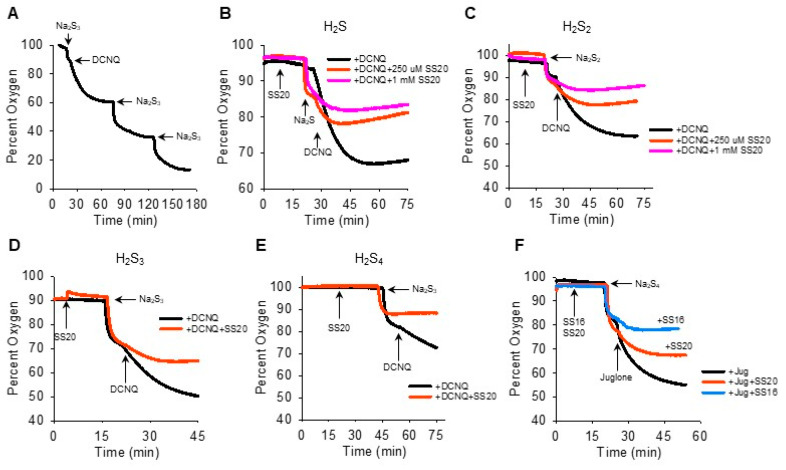
H_2_S scavengers SS20 and SS16 inhibit oxygen consumption in reactions of Na_2_S_1–4_ with DCNQ or juglone. (**A**) Oxygen consumption by 250 μM Na_2_S_3_, 10 μM DCNQ, and subsequent additions of Na_2_S_3_, showing Na_2_S_3_ is the limiting factor. SS20 concentration-dependently decreases oxygen consumption by 10 μM DCNQ and 250 μM Na_2_S (**B**) or 250 μM Na_2_S_2_ (**C**). SS20 (250 μM) inhibits oxygen consumption by 10 μM DCNQ and 250 μM Na_2_S_3_ (**D**) or Na_2_S_4_ (**E**). (**F**) Both SS20 (250 μM) and SS16 (250 μM) inhibit oxygen consumption by 250 μm Na_2_S_4_ and 10 μM juglone.

**Figure 5 antioxidants-13-00619-f005:**
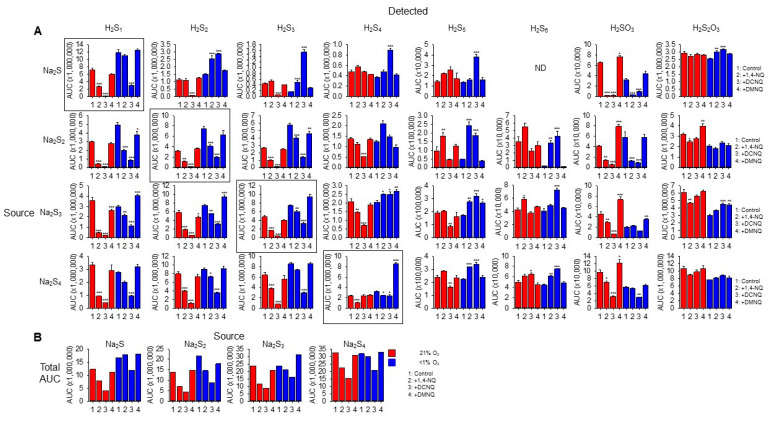
(**A**) LC-MS/MS analysis of sulfur compounds produced by the incubation of 100 μM Na_2_S, Na_2_S_2_, Na_2_S_3_, or Na_2_S_4_ without or with 10 μM 1,4-NQ, DCNQ, or DMNQ for 20 min at 37 °C in either 21% O_2_ (red) or <1% O_2_ (blue). Mean area under the curve (AUC) + SE, n = 3 replicates; *, *p* < 0.05, **, *p* < 0.01, ***, *p* < 0.001 vs. control (no NQ). Boxes show source Na_2_S_n_. (**B**) Total AUC for sulfur species detected as a function of sulfur sources (S_1–4_). Average values include the AUC for HS_4_OH and HS_5_OH from Appendix A. In most instances, there is less total sulfur detected with NQs compared to the control.

**Figure 6 antioxidants-13-00619-f006:**
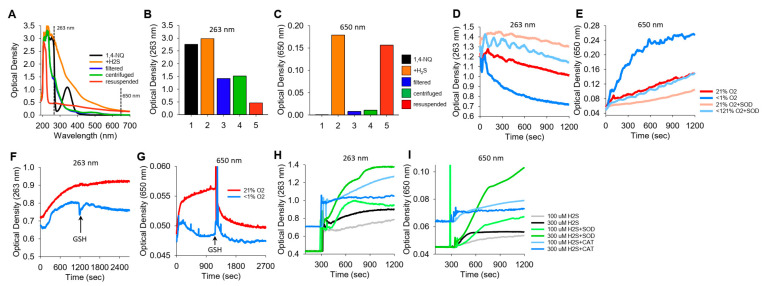
Typical traces showing the optical density (OD) during the formation of soluble and colloidal sulfur (OD_263 nm_) or colloidal S_8_ (OD_650 nm_). (**A**–**C**) OD of 400 μM 1,4-NQ (black), 15 min after adding 400 μM H_2_S to 1,4-NQ (orange), after filtration through a 0.2 μm filter (blue), after 10 min of centrifugation at 14,000 rpm (green), and with centrifuged particles resuspended in PBS (red); (**A**) full spectrum, (**B**,**C**) OD at 263 nm and 650 nm. Colloidal particles from H_2_S oxidation are readily separated by filtration or centrifugation and recovered. (**D**,**E**) Effects of 21% and <1% oxygen and 1 μM SOD on OD_263 nm_ and OD_650 nm_ with 300 μM K_2_S. OD_263 nm_ decreased faster in <1% O_2_ than in 21% O_2_, and SOD appeared to inhibit both responses. Colloid formation followed at OD_650 nm_ was considerably greater in <1% O_2_, and SOD inhibited colloid formation in both 21% and <1% O_2_. (**F**–**I**) Effects of oxygen, GSH, SOD, and catalase on OD during 1,4-NQ oxidation of H_2_S. (**F**,**G**) Incubation of 300 μM Na_2_S and 30 μM 1,4-NQ produced soluble and colloidal S_8_ in 21% O_2_, whereas there was less soluble sulfur and virtually no colloid produced in <1% O_2_. Addition of 1 mM GSH had minimal effects on polysulfides, whereas it rapidly depleted the colloid formed in 21% O_2_. (**H**,**I**) Effects of SOD (0.1 μM) and catalase (Cat, 1 μM) on soluble and colloidal sulfur formation during incubation of 10 μM 1,4-NQ with 100 μM or 300 μM Na_2_S. SOD increased soluble and colloidal sulfur production, whereas catalase was ineffective.

**Figure 7 antioxidants-13-00619-f007:**
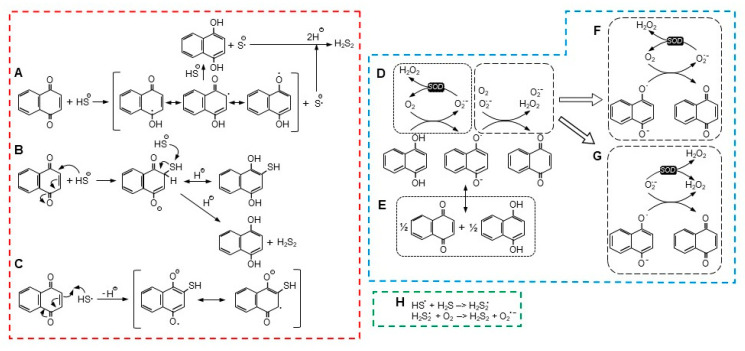
Possible reaction pathways for NQ reduction and H_2_S oxidation ((**A**–**C**), red dashed box), reoxidation of NQ ((**D**–**G**), blue dashed box), and radical formation of H_2_S_2_ ((**H**), green dashed box). See text for details.

**Table 1 antioxidants-13-00619-t001:** Fast and slow components and total oxygen consumed by 250 μM hydropolysulfides without and with 10 μM NQs.

	μM O_2_ Fast	μM O_2_ Slow	μM O_2_ Total
Na_2_S_2_	8.63 ± 1.45	35.88 ± 2.39	44.51 ± 3.85
Na_2_S_3_	20.98 ± 4.33	38.89 ± 3.03	59.87 ± 5.15
Na_2_S_4_	20.23 ± 2.55	32.47 ± 1.41	52.70 ± 3.89
+1,4-NQ	22.68 ± 0.98	63.48 ± 4.90	86.15 ± 4.73
+1,4-NQ	36.37 ± 1.53	65.16 ± 1.24	101.53 ± 2.53
+1,4-NQ	43.87 ± 8.03	50.74 ± 7.56	94.61 ± 3.97
+Juglone	47.97 ± 2.11	59.82 ± 2.37	107.78 ± 4.12
+Juglone	72.30 ± 4.27	41.07 ± 4.08	113.38 ± 8.11
+Juglone	88.79 ± 4.32	18.72 ± 1.90	107.51 ± 5.98
+DCNQ	36.04 ± 3.63	54.97 ± 3.90	91.01 ± 7.47
+DCNQ	59.28 ± 4.14	42.37 ± 3.16	101.65 ± 7.20
+DCNQ	71.39 ± 3.97	28.40 ± 0.23	99.79 ± 3.92
+DMNQ	36.44 ± 4.04	49.32 ± 3.73	85.75 ± 7.74
+DMNQ	69.16 ± 4.05	30.69 ± 2.11	99.84 ± 5.88
+DMNQ	86.82 ± 4.93	5.36 ± 0.22	92.18 ± 4.73

Mean ± SE, n = 3 replicates.

## Data Availability

The original contributions presented in the study are included in the article/Appendix A, further inquiries can be directed to the corresponding author/s.

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
