# Peer review of "Reaction Mechanisms of H2S Oxidation by Naphthoquinones"

_antioxidants, 2024, doi:10.3390/antiox13050619_

Round 1

Reviewer 1 Report

Comments on the  paper entitled  “Reaction mechanism of H2S oxidation by naphtoquinones” submitted to Antioxidants.

      In this original paper Authors studied many aspects of reaction of H2S oxidation by various naphtoquinones. Based on oxygen consumption, polysulfide production by SSP4 as well as LC-MS/MS analysis and optical density for colloidal sulfur formation, Authors collected a lot of results, some of which are included as Supplemental Material. Generally the manuscript is interesting and well-written, however I have some technical corrections and a few comments that the Authors should address.  

Comments:

-          I wonder why naphtoquoinones are not included to keywords – I think they should be obligatory added to the keywords list

-          According the Antioxidants’ Instructions for Authors: “the abstract should be a total of about 200 words maximum” I'm afraid the abstract exceeds 300 words – it should be shortened

-          It is not known what the numbers (1-8) in the abstract mean. I suspect they relate to the graphical abstract but it is not clear when we cannot see it, so I suggest to omit them. Moreover, what does the “(a)” in the line 32 refer to?

-          Naphtoquoinones are abbreviated as NQs, so why in the line 107 “NPs’ is used?

-          All experiments using various NQ described in the paper are performed in buffer solution. It would be valuable if Authors add the results of a few additional experiments performed in the presence of the tissue homogenate which addresses biological conditions.

-          I cannot find the section “Author Contributions”

Minor points:

-          Line 191 – the space between pH and 7.2 is needed

-          Figure 2G – the graph title should be added, like in the case of other graphs (A-F)

-          Figure 2M – the beginning of square bracket for H2S should be added

-          Line 332 – there is the lack of the dot after Fig

-          Line 367 – “ito”?

-          Line 385 – there is the lack of the dot after Fig

-          Lines 442-443 and Line 446 – the same information is repeated

-          Line 456 – the space in 100% is needed

-          Line 479 – O2 instead O2

-          Line 495 – there is the lack of the dot after Fig

-          Line 498 –  the space in than1,4-NQ is needed

-          Figure 4E – in the graph title is H2S4, while on the figure is Na2S3 – is it correct?

-          Figure 4F – the graph title can be added, like in the case of the graphs 4A-E

-          Figure 4 legend – line 537 – SS16 instead SS19

-          Figure 4 legend – line 540 – µM instead µm

-          Line 547 –  probably, there should be dot after Fig. 5A

-          Figure 5 legend – line 557 – HSO4OH  and HSO5OH instead HOS4 and HOS5

-          Line 620 –  probably, there should be dot after Fig. 6A-C

-          Line 631 –  I suggest through instead thru

-          Line 635 –  I suggest than instead thn

-          Line 653 – This sentence should be corrected: “In 21% O OD650 nm SOD produced a further decreased OD650 nm”

-          Line 680 – the space between hydroper-and is needed

-          Line 686 – there should be a space in Fig.1

-          Line 719 – the abbreviation Cys instead cysteine can be used (as in other places)

-          Line 796 – “comproportionation” instead “comproportiontion”

Author Response

The authors thank the reviewers for their thoughtful and constructive comments.  Our responses, indicated with an asterisk are itemized below.

Reviewer #1:

Comments on the  paper entitled  “Reaction mechanism of H2S oxidation by naphtoquinones” submitted to Antioxidants.

      In this original paper Authors studied many aspects of reaction of H2S oxidation by various naphtoquinones. Based on oxygen consumption, polysulfide production by SSP4 as well as LC-MS/MS analysis and optical density for colloidal sulfur formation, Authors collected a lot of results, some of which are included as Supplemental Material. Generally the manuscript is interesting and well-written, however I have some technical corrections and a few comments that the Authors should address. 

Comments:

-          I wonder why naphtoquoinones are not included to keywords – I think they should be obligatory added to the keywords list

* Added per suggestion.

-          According the Antioxidants’ Instructions for Authors: “the abstract should be a total of about 200 words maximum” I'm afraid the abstract exceeds 300 words – it should be shortened

* For some reason the journal editors added the graphical abstract (which was submitted separately) to the text abstract (which is < 200 words).  This has been crossed out in the revised version and highlighted.

-          It is not known what the numbers (1-8) in the abstract mean. I suspect they relate to the graphical abstract but it is not clear when we cannot see it, so I suggest to omit them. Moreover, what does the “(a)” in the line 32 refer to?

*  This text is for the graphical abstract where the numbering is evident when viewed with the figure (also, please see preceding response).

-          Naphtoquoinones are abbreviated as NQs, so why in the line 107 “NPs’ is used?

* Sorry, this was a typo, thank you.

-          All experiments using various NQ described in the paper are performed in buffer solution. It would be valuable if Authors add the results of a few additional experiments performed in the presence of the tissue homogenate which addresses biological conditions.

* This is a very important issue, as the reviewer points out.  We are planning on addressing this in future work using liquid chromatography-tandem mass spectrometry, in conjunction with the fluorophores used here.  However, these are labor (and cost) intensive and we feel these are beyond the scope of the present work.  We ask the reviewer’s indulgence in this.

-          I cannot find the section “Author Contributions”

* The “Author contributions” section was submitted with the ms, We’re not sure what happened.  It is attached below for your review.

KRO, KJC, YG, ZM, EP, AT, JL, KW, ZJ, IK, SMK, PJ Jr, JF, MX, GW, KDS

Conceptualization, KRO;

Methodology, KJC, ZM, GW;

Formal Analysis, KRO, KJC, YG, ZM, EP, AT, JL, KW, ZJ, IK, SMK, PJ Jr, JF, MX, GW, KDS;

Investigation, YG, ZM, EP, AT, JL, KW, ZJ, IK, SMK, GW

Resources, KRO. PJ Jr, MX, KDS

Writing – Original Draft Preparation, KRO;

Writing – Review & Editing, KRO, KJC, YG, ZM, EP, AT, JL, KW, ZJ, IK, SMK, PJ Jr, JF, MX, GW, KDS

Supervision, KRO

Funding Acquisition, KRO, KDS

Minor points:

-          Line 191 – the space between pH and 7.2 is needed

* Corrected, thank you.

-          Figure 2G – the graph title should be added, like in the case of other graphs (A-F)

* We believe it is included.

-          Figure 2M – the beginning of square bracket for H2S should be added

* A corrected figure has been submitted, thank you for pointing this out.

-          Line 332 – there is the lack of the dot after Fig

* Corrected, thank you.

-          Line 367 – “ito”?

* Corrected, thank you.

-          Line 385 – there is the lack of the dot after Fig

* Corrected, thank you.

-          Lines 442-443 and Line 446 – the same information is repeated

* We removed a portion of line 442-443 to correct this, thank you.

-          Line 456 – the space in 100% is needed

* Corrected, thank you.

-          Line 479 – O2 instead O2

* Corrected, thank you.

-          Line 495 – there is the lack of the dot after Fig

* Corrected, thank you.

-          Line 498 –  the space in than1,4-NQ is needed

* Corrected, thank you.

-          Figure 4E – in the graph title is H2S4, while on the figure is Na2S3 – is it correct?

* Na2S4 is correct, thank you.

-          Figure 4F – the graph title can be added, like in the case of the graphs 4A-E

*Added

-          Figure 4 legend – line 537 – SS16 instead SS19

* Corrected, thank you.

-          Figure 4 legend – line 540 – µM instead µm

* Corrected, thank you.

-          Line 547 –  probably, there should be dot after Fig. 5A

*We believe a comma is appropriate here.

-          Figure 5 legend – line 557 – HSO4OH  and HSO5OH instead HOS4 and HOS5

*Corrected to HS4OH and HS5OH

-          Line 620 –  probably, there should be dot after Fig. 6A-C

* We believe a comma is appropriate here

-          Line 631 –  I suggest through instead thru

* Corrected, thank you.

-          Line 635 –  I suggest than instead thn

* Corrected, thank you.

-          Line 653 – This sentence should be corrected: “In 21% O OD650 nm SOD produced a further decreased OD650 nm”

* Corrected to ‘In 21% O2 OD650 nm SOD produced a further decreased in OD650 nm.  Thank you.

-          Line 680 – the space between hydroper-and is needed

* Corrected, thank you.

-          Line 686 – there should be a space in Fig.1

* Corrected, thank you.

-          Line 719 – the abbreviation Cys instead cysteine can be used (as in other places)

*Corrected throughout, thank you.

-          Line 796 – “comproportionation” instead “comproportiontion”

* Corrected, thank you.

Reviewer 2 Report

The manuscript "Reaction Mechanisms of H2S Oxidation by Naphthoquinones" by Olson et al. reports on naphtoquinones' (NQ) catalyzed H2S oxidation evaluating these reactions by monitoring consumption of both H2S and oxygen by NQs under different conditions. This manuscript builds on previous work of the authors, complementing it and extending the range of analysis and derived conclusions.

The manuscript is extremely well written and organised, the "story" flows quite naturally and the reported scientific results are robust with the derived conclusions fully supported (yet cautious, at some stages).

This Reviewer congratulates the authors on the beautiful manuscript they have submitted to Antioxidants and wishes them all the best for their future endeavours.

There are only two Minor comments/typos to report:

1. This Reviewer urges the authors to (re)consider the usage of references in the Abstract (this Reviewer considers this is not standard practice and recommends the removal of these from the Abstract);

2. Section 2.2 title should be "NQs" instead of "NPs", right?

Author Response

The authors thank the reviewers for their thoughtful and constructive comments.  Our responses, indicated with an asterisk are itemized below.

Reviewer #2:

The manuscript "Reaction Mechanisms of H2S Oxidation by Naphthoquinones" by Olson et al. reports on naphtoquinones' (NQ) catalyzed H2S oxidation evaluating these reactions by monitoring consumption of both H2S and oxygen by NQs under different conditions. This manuscript builds on previous work of the authors, complementing it and extending the range of analysis and derived conclusions.

The manuscript is extremely well written and organised, the "story" flows quite naturally and the reported scientific results are robust with the derived conclusions fully supported (yet cautious, at some stages).

This Reviewer congratulates the authors on the beautiful manuscript they have submitted to Antioxidants and wishes them all the best for their future endeavours.

* Thank you so much for your very kind and encouraging words!

  1. This Reviewer urges the authors to (re)consider the usage of references in the Abstract (this Reviewer considers this is not standard practice and recommends the removal of these from the Abstract);

* Thank you for this comment.  For some reason the journal editors added the graphical abstract (which was submitted separately) to the text abstract (which is < 200 words).  This has been crossed out in the revised version and highlighted.

  1. Section 2.2 title should be "NQs" instead of "NPs", right?

*Corrected, thank you.